# Neuropharmacological Effects in Animal Models and HPLC-Phytochemical Profiling of *Byrsonima crassifolia* (L.) Kunth Bark Extracts

**DOI:** 10.3390/molecules28020764

**Published:** 2023-01-12

**Authors:** María de la Cabeza Fernández, Marta Sánchez, Armando Caceres, Irene Iglesias, Maria Pilar Gómez-Serranillos

**Affiliations:** 1Department of Chemistry in Pharmaceutical Science, Faculty of Pharmacy, Complutense University of Madrid, 28040 Madrid, Spain; 2Department of Pharmacology, Pharmacognosy and Botany, Faculty of Pharmacy, Complutense University of Madrid, 28040 Madrid, Spain; 3School of Biological Chemistry, Faculty of Chemical Science and Pharmacy, Universidad de San Carlos, Guatemala City 01012, Guatemala

**Keywords:** *B. Crassifolia*, hole-board test, motor coordination test, CNS activity, analgesic activity, phytochemical analysis

## Abstract

*B. crassifolia* is a species that grows in various areas of Latin America. It was known to be useful for the treatment of different human ailments. The present work evaluated the neuropharmacological and analgesic effects of hydroalcoholic and dichloromethane extracts of *B. crassifolia*. The effect on the central nervous system (CNS) of both extracts obtained from bark, administered by the intraperitoneal route in mice, was evaluated by different tests: spontaneous motor activity, hole-board, motor coordination, pentobarbital induced hypnosis, and rectal temperature. Analgesic activity was evaluated using a hot plate test. Phytochemical analysis was performed by high-performance liquid chromatography (HPLC) using reversed-phase and gradient of elution. The hydroalcoholic extract (dose 0.5 g dry plant/kg weigh) administration caused an important reduction of the head-dipping response in the hole board test. A decrease in spontaneous motor activity test and a disturbance of motor coordination in the rotarod test was observed. The hydroalcoholic extract produced a significant prolongation of pentobarbital induced sleeping time. This extract prevented hot plate test induced nociception. The phytochemical analysis revealed the presence of catechin, epicatechin, and procyanidin B12. Therefore, this study revealed that the hydroalcoholic extract of *B. crassifolia* possesses analgesic and sedative CNS activity.

## 1. Introduction

Since ancient times, the plant kingdom has been an inexhaustible source of pharmacological resources, popularly used to control a wide variety of diseases. An important part of this kingdom is made up of the Ibero-American flora, on which there is extensive ethnopharmacological information, which allows us to glimpse therapeutic alternatives for the treatment of numerous current diseases. *Byrsonima crassifolia*, popularly known as “nanche”, is a tropical tree widely distributed in various areas of Mexico, Central and South America. It usually grows in poor, dry lands often degraded by crops, associated with other trees also called “chaparros”, such as *Curatella americana* [1]. It grows in hot and humid, tropical, or subtropical climates [2]. Its medicinal value has been known and used in these regions since pre-Columbian times. It was used in traditional medicine by various ethnic groups such as the Mixe Indians of Oaxaca and the Zoques, Tzeltal, and Tzotzil of Chiapas (Mexico); Even today it is among the ten most used plants for gastrointestinal disorders in the State of Oaxaca, Mexico [3,4]. The use of the leaves and the bark for the treatment of toothache, vaginitis, diarrhea, bronchitis, and asthma is widespread in different regions of the Yucatan Peninsula, Mexico [5], as well as for the treatment of inflammatory disorders in Central America [6]. The trunk is often used to tan and dye cotton, its wood being very valuable in construction and the manufacture of charcoal [1].

In recent studies, the presence of triglycerides in its composition was verified and it was shown that the oils are made up of long-chain triglycerides, which would mean possible applications in the food and pharmaceutical sectors [7,8].

Numerous studies have been published on the chemical composition of the bark of the tree, isolating triterpene compounds such as β-amyrin, a triterpene acetate, and various dimeric and trimeric proanthocyanidins containing units of (+)-epicatechin and (+)-epigallocatechin with 2S configuration, not very abundant in nature [9]. The bark from samples collected in Oaxaca (Mexico) showed the presence of phenolic compounds such as: catechin and (+)-epicatechin, together with gallic acid, two new procyanidin trimers, and four new dimers [10]. The determination of total polyphenols and flavonoids carried out on bark collected in the State of Pará (Brazil) yielded a content of active ingredients very close to that obtained for the leaves and much higher than that found in the fruits: 38.0 mg in gallic acid equivalents per gram of fresh plant and 11.0 mg of catechin equivalents per gram of fresh plant, respectively [11]. In the samples obtained in Igarapé-aÇu and Bonito (Brazil), the content of active ingredients of this type in the bark was much higher than that obtained in the other parts of the tree analyzed (leaves and fruits): 121 mg of galic acid and 16.2 mg of catechin equivalent per g of dry plant [12].

The ethanolic and aqueous extracts of the bark of *B. crassifolia* have shown good anti-inflammatory activity in the cyclooxygenase inhibition assays, and also antimicrobial activity. On the other hand, extracts of leaves and bark have a concentration-dependent spasmogenic effect on the fundus of the rat stomach in vitro and biphasic effects (concentration dependent inhibition) on the rat jejunum and ileum in vivo [9]. A preliminary study for the scientific validation of the popular use of several of these species confirmed the antidermatophytic effect of the aqueous extract of *B. crassifolia* against all the pathogens studied: *Epidermophyton flocosum*, *Microsporum canis*, *M. gypseum*, *Trichophyton mentagrophytes* and *T. rubrum* [13]. The bark dye is active against Gram negative and Gram positive bacteria such as *Staphylococcus pyogenes*, *Candida krusei*, *C. parapsilosis* and *C. stellatodiea* [14,15]. In subsequent investigations of the ethanolic extract of the bark, an important dose-dependent antimicrobial activity was confirmed against *Bacillus subtilis*, *Escherichia coli*, *Klebsiella pneumoniae*, *Pseudomonas aeruginosa*, *Salmonella typhi*, *Shigella flexneri*, *Staphylococcus aureus*, *S. epidermis*, *Streptococcus pneumoniae* and *Micrococcus luteus* [4].

A study using fruit and seed extracts of *B. crassifolia* collected in Mexico has shown that both enhance the activity of antioxidant enzymes SOD, GSH, GSSG, and CAT in diabetic rats; In addition, they inhibit the formation of glycation end products resulting from non-enzymatic reactions of carbohydrates and oxidized lipids, which are two of the characteristics of type II diabetes. Said inhibition is comparable to that obtained with a standard such as aminoguanidine. The results obtained suggest the prevention of oxidative stress, suppression of liver cell damage, and inhibition of liver dysfunction induced by chronic hyperglycemia, which demonstrates its potential value as an antidiabetic agent. Its possible antihyperglycemic mechanism of action appears to be both pancreatic and extrapancreatic; the latter may be due to sensitization of the insulin receptor in the target organ or inhibition of insulinase activity in the liver and kidney, having ruled out the modulatory effect of glucose absorption at the intestinal level [16].

Various studies carried out on rats treated with ethanolic extracts from the bark and leaves of *B. crassifolia* by intraperitoneal injection demonstrate a dose-dependent decrease in motor activity, mild analgesia, esophthalmos, reversible palpebral ptosis, pale ears, catalepsy, and hypothermia [3,17].

The present study aims to evaluate the neuropharmacological and analgesic activity of extracts from *B. crassifolia* with a view to find the pharmacological rationale for some of the reported and traditional uses of the plant as well as providing comprehensive information of their phytochemical profile. After the realization, a preliminary screening using the Irwin’s test, a series of pharmacological tests were used to explore their activity of on the central nervous system.

## 2. Results and Discussion

### 2.1. Phytochemical Screening

The phytochemical screening indicated the presence of different secondary metabolites in the bark of *Byrsonima crassifolia*: tannins, saponins, flavonoids, anthraquinones and alkaloids.

### 2.2. Extractions Yields

The extraction yield obtained using diclorometane and hydroalcoholic (80:20) solvents for *Byrsonima crassifolia* bark was 2.48% and 30.5%, respectively (Table 1).

### 2.3. Thin Layer Chromatography (TLC)

The extracts prepared from the bark, dichloromethanic (DCM), and hydroalcoholic (HyA) extracts were studied by TLC, in an attempt to reveal the different chemical groups as well as compounds that might be of interest in relation to their pharmacological activity [18].

The analysis for thin layer chromatography revealed the presence of alkaloids, phenolic compounds, and triterpenes in the hydroalcoholic extract, also verifying the existence of tannins in high concentrations. The existence of triterpenes and saponins as major components of the dichloromethane extract was verified.

### 2.4. Identification of Phenolic Compounds by HPLC Analysis

The phenolic profile analysis was carried out by using high-performance liquid chromatography coupled with a photodiode array detector (Figure 1). [19]. Table 2 shows total phenolic compounds detected and quantified in *Byrsonima crassifolia* bark extract, according to stand retention times and UV spectrum. The compounds were assigned to Gallic Ac., Protocatechuic Ac., (+) Catechin, Procyanidin B2, and (−) Epicatechin

#### Validation of the Chromatographic Method

Once the chromatographic conditions had been selected, the method was validated paying attention to the linearity, accuracy, precision, selectivity, quantitation, and stability of standards and samples.

### 2.5. Neuropharmacological Studies

#### 2.5.1. Preliminary Neuropsychopharmacologic Screening

The Irwin’s test is a comprehensive procedure that makes it possible to quantify and collate, in each animal, a wide variety of grossly observable changes produced by drugs (e.g., behavioral, neurological, autonomic, and toxic). After the treatment with drugs, mice were observed every 30 min up to 6 h for studying behavioral changes. Before the treatment with extracts, we conducted the test with standard reference drugs with known effects on CNS [20,21,22,23]. After the administration, the hydroalcoholic extract significantly decreased spontaneous activity and reactivity, in addition to a clear decrease in muscle tone 1/2 h after administration of the extract. The reduction in respiratory rate is also considerable. The depressant effect shown after the administration of 0.5 g dried plant/kg weight of the hydroalcoholic extract was manifested in a decrease of motor activity (spontaneous activity, reactivity, and response to touch), data that would be consistent with the results obtained in previous works [17]. The results obtained in the Irwin test for the dichloromethane extract show a small decrease in spontaneous activity reactivity, and response to touch up to 3 h after the administration of the extract.

This study of the effects on the central nervous system provides strong indications that the hydroalcoholic extract of *B. crassifolia* bark possesses depressant activity, which was manifested throughout the predictive test applied.

#### 2.5.2. Locomotor Activity Measurement

The hydroalcoholic (HyA) and dichloromethane (DCM) extracts caused significant reduction in spontaneous locomotor activity in mice after i.p. administration. In Figure 2, a statistically significant decrease in spontaneous motor activity can be observed at 30, 60, 90, and 120 min after administration for the hydroalcoholic and dichloromethane extracts, with respect to the control and haloperidol (standard reference drug, 6 mg/kg i.p.). The percentage of decrease in spontaneous motor activity was more significant for the hydroalcoholic extract, progressively decreasing until reaching 60.96% compared to the control group, maintaining this effect over time up to 120 min after the administration of the extract.

#### 2.5.3. Hole-Board Test

The results of the hole-board test are reported in Table 3 and Figure 3, showing a significant decrease in the exploratory conduct through the number of head dipping into the hole of the test board. These results were more significant in animals treated with the hydroalcoholic extract than with the dichloromethane extract. Haloperidol (6 mg/kg) was used as a reference or standard substance. The number of times the head is inserted into the holes is counted, not evaluating those performed consecutively in the same hole. At 30 min after the administration of the hydroalcoholic extract there is a significant decrease in curiosity, remaining up to 120 min during the development of the test [23].

#### 2.5.4. Rectal Temperature Test

Body temperature can be interpreted as an index of alteration of various central neurotransmitters, but it also serves to distinguish between total and partial benzodiazepine receptor agonists [24]. The hydroalcoholic extract of the bark administered intraperitoneally at a dose of 0.5 g of dry plant/kg of weight, causes a decrease in rectal temperature in the experimental animals, compared to the control group. No hypothermia was observed with dichloromethane extract.

Rectal temperature shows a decrease at 30 min that is maintained until 120 min, following a behavior like that of the standard group (haloperidol 6 mg/kg). Figure 4 and Table 4 show variations in rectal temperature over time in response to the bark hydroalcoholic and dichloromethanic extracts of *B. crassifolia* (* *p* < 0.05).

#### 2.5.5. Motor Coordination (Rotarod Test)

The first manifestation of CNS depression in the mouse is motor weakness. This weakness is detected by the difficulty or failure in the experimental animal to grip the surface of a rotating cylinder [25]. In the rotarod test, the hydroalcoholic extract of the bark administered intraperitoneally at a dose of 0.5 g of dried plant/kg weight, causes a decrease in motor coordination in the experimental animals, compared to the control group and standard. The decrease in motor coordination is maintained throughout the test and is like that caused by the standard (haloperidol, 6 mg/kg). Table 5 and Figure 5 show the mean values of motor coordination for both extracts, together with the standard deviation and statistical significance (*) *p* < 0.05 with respect to the control group [26].

#### 2.5.6. Induced Sleeping Time Test

In pentobarbital-treated mice, *B. crassifolia* hydroalcoholic extract significantly prolonged the sleeping time at dose 0.5 mg/kg of dry plant/kg of weight [27,28]. The hydroalcoholic extract of the bark administered intraperitoneally causes a decrease in the time of induction to sleep, as well as its duration, and the dichloromethane extract the dose of 1.25 g of dried plant/kg weight does not present statistical significance (Table 6).

### 2.6. Analgesic Activity: Hot Plate Test

The analgesic effect produced by the extracts under study compared with that produced by acetylsalicylic acid (standard), we can observe that both the hydroalcoholic extract and the dichloromethane present a slight analgesic activity 30 min after administration [29,30].

The hot plate test also demonstrated dose-dependent analgesia produced by both hydroalcoholic and dichloromethane extracts of *B. crassifolia* as presented in Table 7 and Figure 6. The differences were observed to be significant as compared to the control and standard.

## 3. Materials and Methods

### 3.1. Plant Material

The bark of *Byrsonima. crassifolia* was collected in its wild growth habitat (Samayac, Suchitepéquez). Samayac, municipality in the department of Suchitepéquez in the southwestern region of the Republic of Guatemala. The botanical identification and authenticated was performed at the herbarium of the University of Valley in Guatemala, (Guatemala) by Dr. Elfriede de Pöll, where an exsiccated sample of the plant with Herbarium registration number 369 was stored under the registration number Herbarium 369. After dried, the material was ground in a mill and stored at room temperature sheltered from light.

### 3.2. Extraction Procedure

The extracts were made by macerating the drug. The plant material (dried powered bark) was subjected to extraction by cold maceration following the method of extraction by increasing polarity, subjecting the drug to cold maceration with an apolar solvent (dichloromethane) for three days and the defatted material was extracted subsequently with a mixture of ethanol/water (80:20). The solvent was evaporated to dryness to obtain solvent free extract. The yield of the dried extract was calculated [31,32].

### 3.3. Phytochemical Screening

The crude hydroalcoholic and dichloromethane extracts of *B. crassifolia* bark was qualitatively tested for the detection of alkaloids, flavonoids, saponins, tannins, and anthraquinones following standard procedures [33].

### 3.4. Phytochemical Screening by Thin Layer Chromatography (TLC)

#### TLC Condition

Phytochemical screening was performed by thin-layer chromatography (TLC) according to [34] using silica gel chromatophils as stationary phase and different mobile phases (*v/v*), with different polarities: toluene/acetone (80:20); ethyl acetate/methanol/water (100:13.5:10); methanol/water (80/20); toluene/chloroform/acetone (40:25:35); ethyl acetate/ethanol/water/ammonia (65:25:9:1) and ethyl acetate/methanol/water (77:15:8) (Table 8). The appropriate selection of suitable mobile phases is essential for the migration and separation of the constituents, which allows us to identify them later. The detection of the separated compounds was initially carried out by observation under UV light, (254 and 365 nm).

### 3.5. HPLC Analysis

The hydroalcoholic extract of *Byrsonima crassifolia* was analyzed (direct injection) by high-performance liquid chromatography coupled to a photodiode array detector. Identification of compounds was performed by using standard compounds (target identification).

#### 3.5.1. Regents and Standards

Solvents of HPLC quality was purchased from Scharlau (Barcelona, Spain). High-purity water was obtained by a Millipore system. Solvents were filtered using 0.20 micron membrane filters (47 mm) (Millipore, Milford, MA, USA) and samples were filtered using 0.20 m membrane filters (13 mm, Millipore, Milford, MA, USA). All solutions were degassed prior to use.

The reference substances, (+)-catechin (purity > 99%), (−)-epicatechin (purity > 99%), procyanidin dimer B2 (purity > 90%), were from Extrasynthèse (Genay, France). Gallic acid (99% purity) and protocathechuic acid. (purity > 98%) were from Sigma-Aldrich (St. Louis, MO, USA).

#### 3.5.2. HPLC Analysis Conditions

The chromatographic system consists of an Agilent 1260 instrument (Agilent Technologies, CA, USA) equipped with a photodiode array detector (190–800 nm). Analytical separation was carried out in a C18 column (4.6 mm × 300 mm × 5 µm, Waters, Cromatografía, S.A Barcelona, Spain) with three mobile solvent phases A: H_2_O/CH_3_COOH (98:2), B: H_2_O/CH_3_COOH/CH3CN (78:2:20), C: CH_3_CN. The elution gradient was developed as shown in Table 9. The sample injection volume was 20 µL and the temperature of the column was fixed at 30 °C. The flow was kept constant at 1.2 mL/min.

#### 3.5.3. Standard Solutions and Calibration Graphs

Reference solutions of decreasing concentrations were obtained by dilution with eluent of the requisite standard solution. These solutions were analyzed and the corresponding peak areas plotted against the concentration of extract injected. The concentrations of the components in the extract of *Byrsonima crassifolia* were calculated from the chromatogram peak areas using the normalization method. The identification of the different compounds was achieved by comparing the tR and absorption spectra obtained for each diluted peak with those obtained for the standards.

Calibration graphs were obtained using six mixtures with all the standards at different concentrations. All samples were prepared and injected in triplicate.

#### 3.5.4. Evaluation of Peak Purity and Linearity

The purity of the eluted peaks was checked with a photodiode array detector (λ = 190–300 nm). The three spectra corresponding to the upslope, apex, and downslope of each peak were computer normalized and superimposed. Peaks were considered pure when there was an exact coincidence among the three spectra (match factor ≥ 99.5). The linearity of the detector responses for the prepared standards was assessed by means of a linear regression analysis regarding to the amounts of each standard (measures in μg) introduced in the loop of the chromatographic system and the area of the corresponding peak on the chromatogram.

### 3.6. Test Animals

For the pharmacological tests, non-blood male and female mice, albino variety, CD/1 Swiss strain from Charles River (CRIFFA, Barcelona, Spain) were used (weight between 23 ± 2 g). Mice were divided into four groups: control, standard, HyA, and DCM extract (*n* = 5). The animals were maintained on standard chow and water ad libitum, until they were used for study. In all cases, a period of at least four days passed before their use and at the time of experimentation. The mice were kept in groups of five per cage with an ambient temperature of 22 °C, light/dark cycles of 12 h, and food and water were provided ad libitum (standard diet, Interfauna, Barcelona, Spain). All animal care and experimental protocols conformed to the European Union Guidelines for the Care and the Use of Laboratory Animals (European Union Directive 2010/63/EU).

### 3.7. Drugs and Dosage

The following drugs and dosage were used: carbachol HCl (Sigma) 2 mg/kg; atropine sulfate (Sigma) 2.5 mg/kg; scopolamine HBr (Sigma) 2 mg/kg; epinephrine bitartrate (Sigma) 1 mg/kg; amphetamine bitartrate 5 mg/kg; lidocaine HCl (Sigma) 40 mg/kg; acetyl salycilic acid,100 mg/kg; morphine HCL, 10 mg/kg; haloperidol (Syntex latino, SA) 1 and 6 mg/kg. Each drug was dissolved in isotonic saline solution (0.9% NaCl) just before use, and the resulting solution was administered by intraperitoneal (i.p.) injection of 0.2 mL. The hydroalcoholic extract of the bark was administered intraperitoneally at the dose of 0.5 g dried plant/kg weight, and dichloromethane extract at 1.25 g dried plant/kg weight.

### 3.8. Initial Pharmacological Screening

Doses of the test material were administered to groups of mice (*n* = 5) and grossly observable behavioral effects were observed and quantified as was previously described by Irwin and Mathiasen et al. [20,35]. The Irwin’s test is a comprehensive procedure that makes it possible to quantify and collate, in each animal, a wide variety of grossly observable changes produced by drugs (e.g., behavioral, neurological, autonomic, and toxic). After the treatment with drugs, mice were observed every 30 min up to 6 h for studying behavioral changes. Before the treatment with extracts, we conducted the test with standard reference drugs that has very known effects on CNS: atropine, carcbachol, scopolamine, epinephrine, amphetamine, lidocaine, and morphine.

### 3.9. Neuropharmacological Studies

#### 3.9.1. Spontaneous Motor Activity Test

The spontaneous locomotor activity of mice was investigated in the open field test. A PANLAB Actimeter (model 0602) was used to measure the spontaneous motor activity of animals. This activity cage contains photoelectric cells that are sensitive to any motion within it placing the animals in two cavities connected to two photoelectric plates. Using a digital counter, the number of movements made by the animals was automatically recorded.

The group to which we administered physiological saline was placed in one of the boxes. In another of the boxes, the batch of mice to which the extract under study has been administered was placed. Measurements were made every 30 min for 120 min. The movements of the animals to which the sample has been administered were compared with those that have only been treated with physiological saline. Haloperidol (1 mg/kg i.p.) was used as a standard reference drug.

#### 3.9.2. Hole-board Test

The measurement of this parameter was carried out according to the method previously proposed by Boissier and Simon [36]. For this study, a specific device is available, a model PTR 16 CiBERTEC, which consists of a plastic sheet of 35 cm on each side and 5 cm thick, with 16 holes of 28 mm in diameter, regularly arranged, and with transparent plastic walls, thus the experimental animals cannot escape.

Each hole is traversed, 12 mm below the top of the plate, by a photoelectric cell. All the photoelectric cells are connected to a monitor that records the movements on a digital counter. Each time the animal introduces its head into the hole, the monitor counts this movement.

Mice should be placed on the plate at a certain time, otherwise the percentage of error would be significant. This head-dipping behavior is known as an indicator of the exploratory rate of mice in the apparatus and correlates with anxiety levels. Haloperidol (1 mg/kg) was used as a standard reference drug. Exploratory behavior was recorded every 5 min for an hour [37].

#### 3.9.3. Rectal Temperature Test

Rectal temperature was measured with a digital thermometer (PANLAB model TMP812, resolution 0.1 degrees). Temperature was recorded just before (T0) the administration of the extracts and at 30, 60, 90, and 120 min after administration.

#### 3.9.4. Motor Coordination Test (Rotarod Test)

The first manifestation of CNS depression in the mouse is motor weakness. This weakness is detected by the difficulty or failure in the experimental animal to grip the surface of a rotating cylinder [27].

The rotarod test was employed to measure the motor coordination and resistance following the technique described by Dunham et al. [38]. The mice were assessed for motor coordination and balance using a rotarod apparatus (LE 8500 Letica Scientific Instrument, Barcelona, Spain). The time the animal remains on the cylinder without falling is counted. Mice were initially trained to remain on the rotarod apparatus. The next day, mice were placed on the rotating rod that accelerated smoothly from 4 to 16 rpm over a period of 2 min. The length of time they could maintain their balance on the turntable against the movement’s strength was recorded. Then, the extract or vehicle was injected and after 30 min the animals were placed on the rotarod again, recording the values at 30 (t30), 60 (t60), 90 (t90), and 120 (t120) min. The animals assigned to each experimental group received the extracts, vehicle, or standard (haloperidol), each mouse was placed on the rotarod to evaluate the time of permanency (spent) on the rotating bar [39].

#### 3.9.5. Pentobarbital-Induced Sleep Test

Sleep time was evaluated in pentobarbital-induced mouse sleep model. The onset of sleep is the time that animals stayed immobile and lost their righting reflex. The sleep latency was recorded as the time between administration of pentobarbital and onset of sleep [40].

The hydroalcoholic (HyA) and dichloromethane extracts (DCM) of *B. crassifolia* were intraperitoneally (ip) administrated to mice 30 min before injection of sodium pen-tobarbital (60 mg/kg, ip). The time between injection and loss of the righting reflex induced by pentobarbital was considered the latency time. Furthermore, sleeping time was considered as the difference between the time of loss and recovery of the righting reflex [41]. Control group was i.p. treated with the vehicle alone (saline solutions) to determine the duration of hypnosis induced by pentobarbital.

#### 3.9.6. Analgesic Activity. Hot Plate Test

The method we followed to verify the existence of analgesic activity of the species under study was the hot plate test [42] since it allows us to study both the central mechanism and peripheral mechanism analgesia. The method as modified by Connor et al. [43] was employed for this test. The mice were distributed in four batches of five animals each. The temperature was maintained at about 54 ± 0.5 °C. The assay endpoint was the first instance of hind paw-licking. A hot plate mod. Ugo Basil, Italy; Socrel DS-37 was used, which consists of a metal surface which, by means of a system of electrical resistances, is maintained at the desired temperature. It also has a cylinder made of a transparent plastic material that allows the animal to be confined to the heated surface and to be always observed. It also has a chronometer that allows registering with an accuracy of 0.1 s the time of permanence under the nociceptive stimulus. A 60-sec cut-off was imposed to prevent tissue damage. Each mouse was tested twice before drug, saline, or standard administration. The test was repeated for each mouse 30, 60, 90, and 180 min after extracts, standard drug, or normal saline administration. The time at which the peak effects occurred was selected.

For each batch of animals, the average of the reaction times (t after) was calculated at each time and with this value and the one obtained before the administration of the extracts, the percentage of analgesia was calculated.

### 3.10. Statistical Analysis

Statistical differences between values of the assays were obtained by analysis of variance (ANOVA) to the significance level of 5% and indexes of correlation and regression were determined into the program Statgrafics Plus4.1. The differences were considered significant at * *p* < 0.05.

## 4. Conclusions

In the present work, the effects on central nervous system of *B. crassifolia* extracts were analyzed in several behavioral models. The present research reveals the presence the phenolic compounds and evaluates the neuropharmacological profile activities of the dichloromethane and hydroalcoholic extracts from the bark of *Byrsonima crassifolia.* The obtained results reveal the richness of the hydroalcoholic and dichloromethane extracts of *Byrsonima crassifolia* in tannins and phenolic acids.

The spontaneous locomotor activity test was used as a general parameter in investigating the central action of a drug. A decrease in locomotor activity means that the drug has a depressing effect on the CNS in the study animal, in this work, a decrease in mobility capacity can be seen after the administration of the hydroalcoholic extract. The validation of the anxiety was carried out by measuring external signs, through the hole-board test. The exploration capacity might be considered an index of anxiety, although it is difficult to separate it from motor activity. The results show a significant decrease in exploratory conduct in the mice caused by the hydroalcoholic extract. Upon testing the muscle relaxant activity, it was found to produce decreased effects on motor coordination. Furthermore, the hydroalcoholic extract of *Byrsonima crassifolia* produced significant variations on the behavior of treated animals and increased the pentobarbital-induced sleeping time. This extract showed reduced spontaneous motor activity, the results show a significant decrease in exploratory conduct in the mice caused by the hydroalcoholic extract and the caused significant hypothermia can be interpreted as an alteration of various central neurotransmitters, probably due to the presence of (-) epicatechin and their derivatives, procyanidin B2 and gallic acid in its composition.

The presence of phenolic products as major chemical constituents in the extracts may demonstrate the contribution of these compounds for activity at the level of the central nervous system observed in this study. From the above experiments, it could be concluded that hydroalcoholic extract of *B. crassifolia* contains significant neuropharmacological activities. To elucidate the exact mechanism action and bioactive compounds responsible for the neuropharmacological and analgesic activities of this plant extract, further pharmacological studies must be performed.

## Figures and Tables

**Figure 1 molecules-28-00764-f001:**
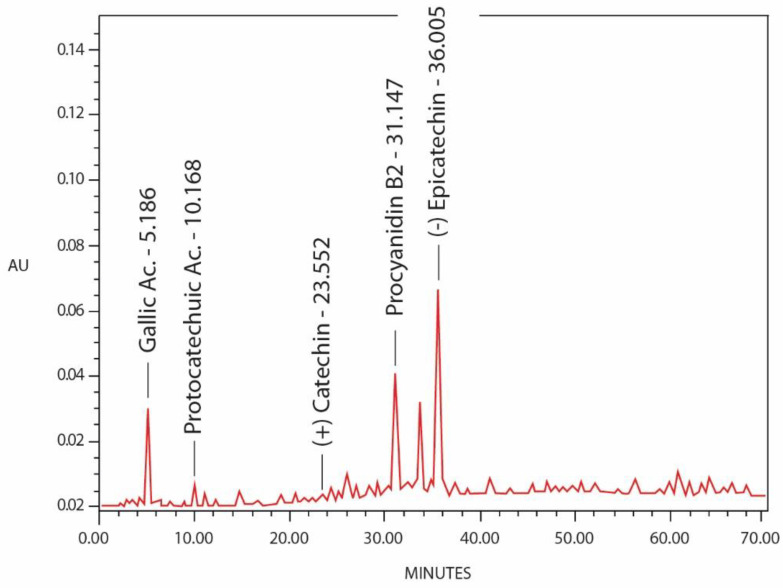
HPLC chromatographic profile of phenolic compounds detected in *Byrsonima crassifolia* extract (EtOH: H_2_O 8:2 *v/v*), acquired at 300 nm.

**Figure 2 molecules-28-00764-f002:**
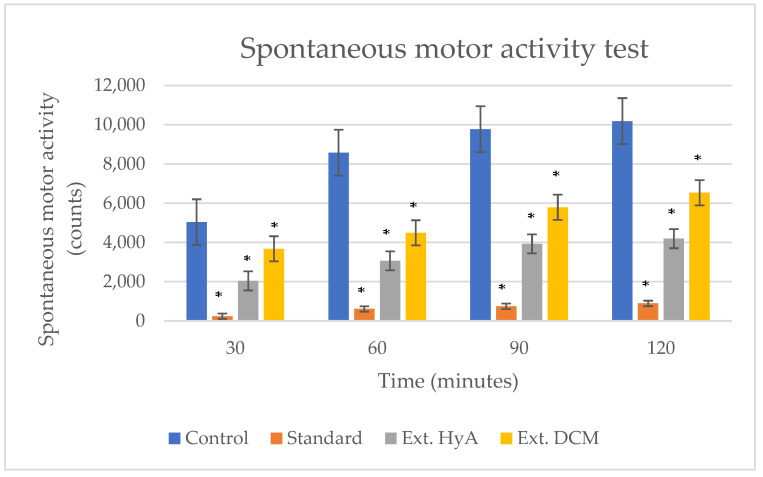
Effects of *Byrsonima crassifolia* on spontaneous motor activity. Haloperidol (6 mg/kg i.p.) was used as a standard reference drug. * *p* < 0.05 significant compared to control. Ext. HyA: hydroalcoholic extrac; Ext. DCM: dichloromethane extract.

**Figure 3 molecules-28-00764-f003:**
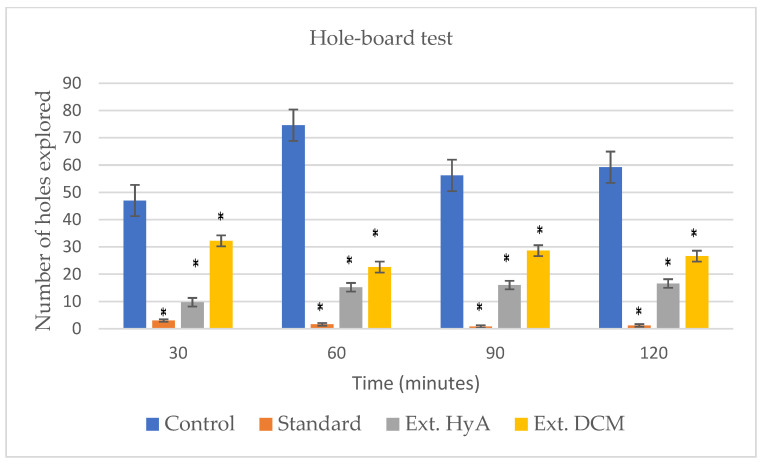
Effects of hydroalcoholic and dichloromethanolic extracts of *B. crassifolia* bark on hole board test. * *p* < 0.05, significant as compared to control.

**Figure 4 molecules-28-00764-f004:**
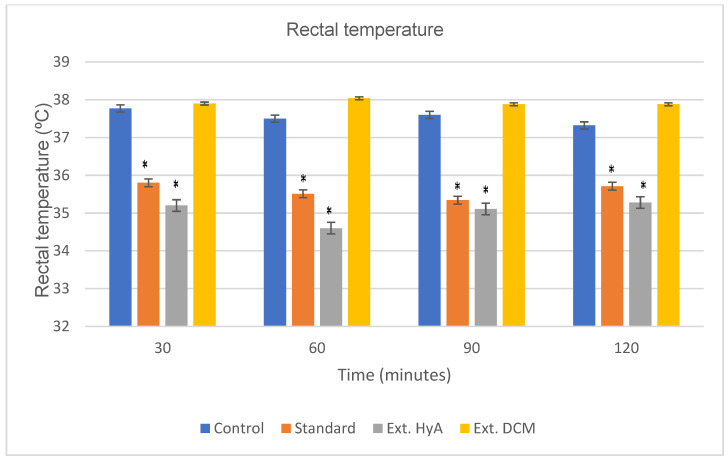
Effects of *B. crassifolia* bark extracts on normal body temperature of mice (temperature values expressed as mean ± SEM, from 5 animals in each group). * *p* < 0.05 significant as compared to control group. Ext. HyA: hydroalcoholic extract; Ext. DCM: dichloromethane extract.

**Figure 5 molecules-28-00764-f005:**
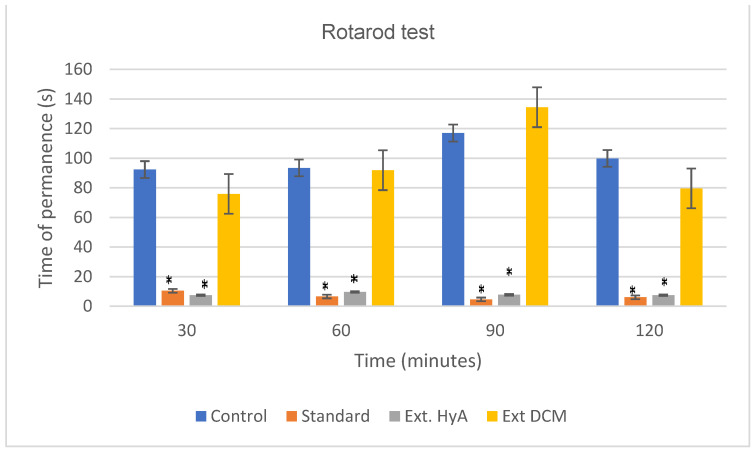
Effect of *Byrsonima crassifolia* extracts administration in mice on Rotarod test. * *p* < 0.05 significant as compared to control group. Ext. HyA: hydroalcoholic extract; Ext. DCM: dichloromethane extract.

**Figure 6 molecules-28-00764-f006:**
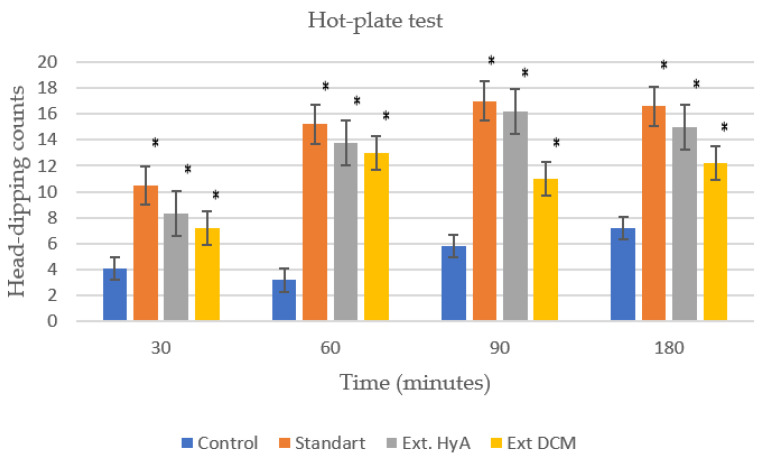
Analgesic activity of the dichloromethane and hydroalcoholic extracts of *Byrsonima crassifolia* as compared to control group and standard (acetyl salycilic acid, 100 mg/kg). Ext. HyA hydroalcoholic extract; Ext. DCM dichloromethane extract. * *p* < 0.05 significantly different as compared to control group.

**Table 1 molecules-28-00764-t001:** Plant extract yields obtained of the bark of *B. crassifolia* with dichloromethane and ethanol-water (80:20) extract.

Extract	Yield (%)
Dichloromethane (DCM)	2.48
Hhydroalcoholic (HyA)	30.5

**Table 2 molecules-28-00764-t002:** Retention time and content of the main identified phenolic compounds in *B. crassifolia* hydroalcoholic extract (quantitative data refer to the extract as it is). Values are the means of three determinations ± SD.

Phenols	Retention Time (*t*_R_ ± SD) (min)	Concentration (µg/100 mg)
Gallic Ac.	5.186 ± 0.07	0.205 ± 0.08
Protocathechuic Ac.	10.168 ± 0.10	0.092 ± 0.11
(+) Catechin	23.552 ± 0.14	0.101 ± 0.16
Procyanidin B2	31.147 ± 0.18	1.200 ± 0.21
(−) Epicatechin	36.006 ± 0.09	0.405 ± 0.07

**Table 3 molecules-28-00764-t003:** Number of holes crossed by the mice of different groups in the hole-board test. Values are mean ± SEM (*n* = 5); * *p* < 0.05, significant compared to control. Ext. HyA: hydroalcoholic extract; Ext. DCM: dichloromethane extract.

Time (min)	Control	Standard	Ext. HyA	Ext. DCM
30	47 ± 2.43	3 ± 1.58 (*)	9.70 ± 6.26 (*)	32.20 ± 6.26 (*)
60	74.60 ± 1.70	1.60 ± 1.80 (*)	15.20 ± 1.270 (*)	22.60 ± 5.60 (*)
90	56.20 ± 1.90	0.80 ± 1.70 (*)	16 ± 8.40 (*)	23.60 ± 1.00 (*)
120	59.20 ± 1.99	1.20 ± 1.30 (*)	16.5 ± 8.40 (*)	26.6 ± 12.02 (*)

**Table 4 molecules-28-00764-t004:** Temperature values obtained for both extracts are reflected, along with the standard deviation and statistical significance (*) *p* < 0.05 with respect to the control group. Ext. HyA: hydroalcoholic extract; Ext. DCM: dichloromethane extract.

Time (min)	Control	Standard	Ext HyA	Ext DCM
30	37.77 ± 0.22	35.88 ± 1.02 (*)	34.27 ± 0.78 (*)	37.92 ± 0.40
60	37.58 ± 0.34	35.51 ± 1 (*)	35.6 ± 0.72 (*)	38.04 ± 0.13
90	37.65 ± 0.39	35.34 ± 0.41 (*)	35.11 ± 0.55 (*)	37.88 ± 0.19
120	37.32 ± 0.44	35.71 ± 0.28 (*)	35.28 ± 0.90 (*)	37.88 ± 0.21

**Table 5 molecules-28-00764-t005:** Effects of *B. crassifolia* bark extracts on motor coordination. Values are mean ± S,E,M, (*) *p* < 0.05 significant as compared to control group. Ext. HyA: hydroalcoholic extract; Ext. DCM: dichloromethane extract.

Time (min)	Control	Standard	Ext HyA	Ext DCM
30	92.28 ± 66	10.42 ± 5.70 (*)	7.42 ± 2.37 (*)	75.85 ± 65.70
60	93.42 ± 75.10	6.57 ± 1.81 (*)	9.57 ± 5.30 (*)	91.85 ± 67.10
90	117 ± 74.20	4.57 ± 0.70 (*)	7.71 ± 2.70 (*)	134.4 ± 73.60
120	99.85 ± 79.59	6 ± 2.23 (*)	7.42 ± 3.86 (*)	79.57 ± 50.40

**Table 6 molecules-28-00764-t006:** Effects of *B. crassifolia* on latency and the duration of the pentobarbital induced hypnosis. Extracts were injected i.p. 30 min before pentobarbital. Each value represents the mean ± SEM of results from five mice. Statistical differences **p* < 0.05 versus control group.

	Pentobarbital (60 mg/kg i.p.)	
Treatment	Latency Time (min)	Sleeping Time (min)
Saline solution	4 ± 0.50	60.09 ± 1.40
Hydroalcoholic extract	2.10 ± 0.70 *	115.35 ± 1.70 *
Dichloromethane extract	6 ± 1.20	75.60 ± 3.20

**Table 7 molecules-28-00764-t007:** Results obtained for the hot plate test * *p* < 0.05 significantly different as compared to control group. Ext. HyA: hydroalcoholic extract; Ext. DCM: dichloromethane extract.

Time (min)	Control	Standard	Ext HyA	Ext DCM
30	4.10 ± 1.20	10.50 ± 1 (*)	8.30± 2.58 (*)	7.20± 1.10 (*)
60	3.20 ± 2	15.20± 0.59 (*)	13.80± 1.75 (*)	13± 2.43 (*)
90	5.80 ± 2.10	17 ± 2.20 (*)	16.20 ± 2.35 (*)	11 ± 2.62 (*)
180	7.20 ± 1.50	16.60 ± 1.63 (*)	15 ± 1.48 (*)	12.20 ± 3.20 (*)

**Table 8 molecules-28-00764-t008:** Chemical groups, mobile phases, and reagents used in TLC for the characterization of *Byrsonima crassifolia* extracts.

Chemical Group	Mobile Phase (*v/v*)	Detection Reagents
Alkaloids	Toluene/Acetone (80:20)	Mayer Dragendorff
Flavonoids	Ethyl Acetate/Methanol/Water (100:13.5:10)	NEU
Saponins	Methanol/Water (80/20)	Sulfuric vanillin
Tannins	Toluene/Chloroform/Acetone (40:25:35)	Ferric salts Cl_3_Fe.
Cumarinas	Ethyl Acetate/Ethanoll/Water/Ammonia (65:25:9:1)	Bornträger
Triterpenes	Ethyl Acetate/Methanol/Water (77:15:8)	Ehrlich

**Table 9 molecules-28-00764-t009:** Gradual variation of the mobile phase composition during elution (For conditions, see the text).

Step	Time (min)	Mobile Phase (%)		
		A	B	C
1	0.00	100.0	0.0	0.0
2	55.00	20.0	80.0	0.0
3	57.00	10.0	90.0	0.0
4	70.00	10.0	90.0	0.0
5	80.00	5.0	95.0	0.0
6	90.00	0.0	100.0	0.0
7	100.00	0.0	20.0	80.0
8	105.00	0.0	0.0	100.0
9	110.00	100.0	0.0	0.0
10	120.00	100.0	0.0	0.0

## Data Availability

Not applicable.

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
