# Peer review of "Neuropharmacological Effects in Animal Models and HPLC-Phytochemical Profiling of Byrsonima crassifolia (L.) Kunth Bark Extracts"

_molecules, 2023, doi:10.3390/molecules28020764_

Round 1
Reviewer 1 Report
Extracts from Byrsonima crassifolia have been studied for a long time and for various activities. Extracts from this plant have a fairly wide range of biological activity: anti-inflammatory, antioxidant, antimicrobial and others. This work is a logical continuation of the work of the authors on the study of the neuropharmacological potential of extracts of ethno-medicinal plants from Guatemala.
1. Why were the extracts immediately applied to animals, while there are no data on their study on brain cells in vitro?
2. What is the LD50 dose? Or were the extracts not toxic to animals at the highest doses tested?
3. Deviations should be indicated on the graphs.
4. Why is a different dose of hydroalcoholic extract used in the sleep test?
5. Why is the dose of haloperidol 1 mg/kg used in the locomotor test, and 6 mg/kg in others (rectal temperature, Holebord, Rotarod)?
There are also errors and typos in the text, for example:
- line 108 - secondary
- line 171 of another Byrsonima crassifolia font
- Table 1 - peak labels on the chromatogram are hard to read
- fix the inscription to table 5, as well as the signature of the DCM statement in the table header
- line 229 - significantly
- line 420 - diclomethane extracts
Author Response
We strongly appreciate the detailed valuable comments of the referees on our manuscript “Neuropharmacological effects in animal models and HPLC-Phytochemical Profiling of Byrsonima crassifolia (L.) Kunth bark extracts”. The suggestions have been quite helpful for us and we have incorporated them in the revised review paper for improving the quality of this work.
As below, we have tried to respond, point by point, all the questions raised by the Reviewer. We hope that everything is all right.

Reviewer 2 Report
Line 18; “central nervous system (CNS)”, and it allows to use this abbreviation at the end of the text - Line 29.
Lines 40-42; Text is not obvious ; Line 40; “…in poor, dry lands often degraded by crops…” and line 41; “…grows in hot and humid, tropical…”
Line 67-68; Should be; “…121 mg of gallic acid and 16.2 mg of catechin equivalent per g of dry plant.” (?) without “respectively”.
Line 70; should be; “…cyclooxygenase inhibition assays, and also antimicrobial activity.”
Line 72-73; Is; “Increased concentration followed by inhibition” – it should be “concentration dependent inhibition “ (?)
Line 78 and 79; “C. parapsilosis, C. stellatodiea [14,15]”
Line 112; should be „dichloromethane” and “ethanol-water (80:20) extract”
Abbreviations should be therefore; DCM and EW80 or something like that. If it will stay “HyA” it should be consistently used in the text and in the Tables; see caption for Figure 1, and caption for Figure 2 (where this abbreviations are; “Ext. DCM” and “Ext. HyA” – the same in the histogram on the Figure 2 and in the Table 3., Table 4, and in Table 5, where there is different abbreviation used; “Ext Hyd” and “Ext Dic” and in the caption abbreviation is different. – Please decide which abbreviations will be used, and use this in the whole Manuscript.
In the Table 2 caption is not properly used; This Table illustrates not only retention times of the phenolic compounds. If concentration is listed please give us information was it concentration per g (per 100 g) of the extract (or other unit). Is "mg" a proper unit as we see so low amounts – maybe micrograms will be better, however, we don’t know if these micrograms are per g of extract, per 100 g of extract (or substance, dry weight or wet weight). It is confusing.
These results should be presented with standard deviation. Validation should be performed if quantitative analysis is presented. Calibration curves for standards should be given (area is not necessary to be presented, and it is only for calculations).
Table 2; It is; “Ac. Gallic”, “Ac. protocatechical” - It should be; “gallic acid, protocatechuic acid”. The same problem in Line 431.
Page 4; Figure 1 should be corrected – additional window is not very informative. The peak of the (-) epicatechin is not clearly visible.
Results in the Tables; the same points of decimals should be given.
Table 6 – in the column entitled “treatment” we have once “B. Crassifolia”, and next line “Brysonima crassifolia” – Is it necessary to repeat the plant name ? And in different ways (?) In this column only abbreviations of the names of extracts used should be listed.
In the Materials and Methods section subsection “Reagents and Chemicals” should be given containing list of reagents, standard substances – their producers and purity.
How was the extract prepared before HPLC analysis? Was it purified/fractionated ? Especially it is interesting and important, because HPLC analysis was done for quantitative estimation of the phenolic content (see Table 2).
Page 3 – TLC analysis – results are not presented. In the Page 10 the description of the TLC methods is not adequate – we have no idea for which groups of detected “alkaloids, triterpenes, tannins, saponins “ (lines 126-129) which mobile phase and detection reagents were used.
For all of the listed mobile phases we should have information are they compositions prepared as v/v or as m/m. Sections 3.4.1 and 2.3 should be rewritten properly to be more informative.
Line 431; should be “hydroalcoholic”
Plant systematic names should be in italics – see references 3,4,6,7,8,10, 22,33,41 and 43
Line 548; In the Ref. 38 journal abbreviation should be given.
Author Response

(The authors gave the same response as above.)

Round 2
Reviewer 1 Report
After reading the answers of the authors, I still believe that in further research it is first necessary to conduct experiments on cell cultures, and then on animals. Taking into account all ethical standards of work with laboratory animals.